# Climate change and its impact on wheat distribution in semi-arid ecosystems: A case study from the Sultanate of Oman

Khalifa M. Al-Kindi[1]*, Ali Hussain Al-Lawati[2]*

1 UNESCO Chair of Aflaj Studies, Archaeohydrology, University of Nizwa, Nizwa, Sultanate of Oman,
2 Natural and Medical Sciences Research Center, University of Nizwa, Nizwa, Sultanate of Oman

* alkindi.k@unizwa.edu.om (KMA-K); ali.allawati@unizwa.edu.om (AHA-L)

## Abstract

Climate change, characterised by long-term shifts in temperature, precipitation, and extreme weather events, poses significant challenges to agricultural sustainability. This study aims to mitigate the impact of climate change on wheat production in Oman by identifying optimal cultivation areas for four temporal periods. Utilising the maximum entropy (MaxEnt) model, the study assessed the suitability of wheat cultivation across four periods: 1970–2020 (reference period), 2021–2040, 2041–2060 and 2061–2080. The model considered environmental variables, such as temperature and precipitation, to predict wheat distribution for the present and future climate scenarios. The MaxEnt model demonstrated robust predictive performance, with area under curve (AUC) scores consistently above 0.8 across all periods. The model achieved an AUC of 0.82 for the reference period (1970–2020) and accurately identified the regions suitable for wheat cultivation. The AUC for the immediate future (2021–2040) decreased marginally to 0.81, reflecting potential shifts in environmental conditions that might influence wheat distribution, and returned to 0.82 for the 2041–2060 period, indicating the model's resilience in predicting wheat suitability despite the projected climate change impacts. Notably, the AUC increased to 0.83 for the 2061–2080 period, suggesting that wheat distribution patterns might become more distinct under future climate scenarios or that the environmental variables driving the model gain greater significance as climate change intensifies. These results highlight the effectiveness of the MaxEnt model in identifying suitable wheat cultivation areas in varying climate conditions. The results provide critical insights into Oman's long-term agricultural planning and sustainable practices. Given the historical wheat cultivation in different regions of Oman, it is crucial to identify optimal areas for future production under climate change to ensure food security and support strategic decision-making. This study emphasises the importance of integrating predictive modelling into agricultural planning and calls for further research to refine strategies for climate-resilient wheat production.

**Data availability statement:** Data provided as supporting information Please see File 1, 2, 3, 4 in S1 File and S2 File in the revised revised and clean version.

**Funding:** The author(s) received no specific funding for this work.

**Competing interests:** The authors have declared that no competing interests exist.

## 1. Introduction

The Sultanate of Oman in the south-eastern Arabian Peninsula is rich in natural resources, including oil and natural gas, integral to the economic system [1]. Its diverse geography supports extensive biodiversity, with landscapes ranging from mountains and deserts to a long coastline [2]. Appreciation of Oman's geography and climate is vital for agricultural studies and assessing the impacts of climate change [3,4].

Climate change refers to long-term alterations in weather patterns, including temperature, precipitation and extreme events, typically observed over 30 years. It is driven primarily by greenhouse gas (GHG) emissions such as carbon dioxide, methane, nitrous oxides and chlorofluorocarbons [5]. While natural systems contribute to climate change, the dominant cause is human activities [6,7]. Its impacts span environmental and economic domains, significantly affecting food production, especially in developing countries [8–10]. Studies revealed that climate change threatens food security by affecting agricultural systems. It reduces livestock productivity, increases mortality and degrades soil health through erosion, nutrient runoff and loss of soil quality. Climate change affects plants by altering their phenological development, productivity and biomass, reducing the chilling hours for perennials, and changing their growing degree days [11]. Bajwa et al. [12], highlighted the detrimental impact of pest infestations on wheat production, underscoring the compounded challenges facing agriculture.

Despite Oman's dependence on agriculture for its food production and its vulnerability to climate change, few studies have explored the spatial distribution of wheat and its interaction with bioclimatic factors. Previous studies overlooked the unique climate zones of Oman and the fundamental challenges impacting its agricultural sector [13,14]. Generalisations from global studies on wheat modelling failed to capture the nuanced bioclimatic conditions in Oman [15]. The absence of research tailored to the diverse climate zones of Oman — coastal, mountainous and desert — presents an apparent lack of clarity on the impacts of varying climates on wheat cultivation [16]. Additionally, limited investigations linking bioclimatic variables to wheat distribution hindered the development of localised insights crucial for effective agricultural planning [17].

Factors such as temperature and precipitation, which significantly influence the spatial distribution of wheat, are particularly important in regions such as Oman, where water scarcity and climate variability are pressing challenges [18]. The lack of predictive modelling for future wheat distribution in Oman represents a substantial research gap [19]. Understanding the future wheat distribution under climate change scenarios is crucial for developing adaptive agricultural strategies. Without such projections, decision-makers and stakeholders lack the critical data for implementing robust and sustainable farming practices [20,21].

Targeted research integrating tools such as geographical information systems, remote sensing and predictive modelling techniques like maximum entropy (MaxEnt) with state-of-the-art climate projections from the Coupled Model Intercomparison

Project Phase 6 (CMIP6) framework to evaluate the impact of climate change on wheat distribution in Oman are pivotal to bridge the gap [22,23]. The MaxEnt model is particularly effective for understanding how climate change influences wheat cultivation potential [24–26]. For instance, Zhao et al. [27], applied MaxEnt to evaluate the potential wheat distribution under 1.5°C and 2°C global warming scenarios and found that climate change would significantly affect planting suitability, with over half of the areas experiencing shifts due to rising temperatures. Similar studies in Oman could address current knowledge deficits, offering novel and valuable insights into sustainable agriculture and climate adaptation [28,29].

The primary objective of this investigation is to mitigate the effects of climate change on wheat production in Oman by utilising the MaxEnt model to identify the most favourable cultivation areas across four temporal periods: 1970–2020, 2021–2040, 2041–2060 and 2061–2080. The significance of this study is its ability to help the government and investors make informed decisions regarding suitable locations for wheat cultivation. Despite the awareness of wheat farming obstacles in Oman [30], a comprehensive assessment of the most favourable areas for wheat cultivation under shifting climate conditions remains absent. Considering the historical wheat cultivation in various Omani regions, it is crucial to determine the most suitable areas for future production amidst climate change [31]. In-depth investigations in this area are necessary to inform strategic decision-making processes. The study outcomes enhance comprehension of how climate change affects agricultural sustainability and wheat production in semi-arid ecosystems, which is pivotal for creating adaptive strategies to ensure food security.

## 2. Materials and methods

### 2.1. Study area

Oman is located on the southeastern edge of the Arabian Peninsula, bordered by the Arabian Sea to the southeast, the Sea of Oman to the northeast, and the United Arab Emirates and Saudi Arabia to the west. Oman also shares a maritime boundary with Iran, placing it strategically near the Strait of Hormuz to the north. It has significantly diverse geographical features comprising vast mountain ranges, expansive deserts, and a long coastline. The Al Hajar mountain range runs through the northern part of the country (Fig 1), while the Rub' al Khali, or Empty Quarter, one of the world's largest uninterrupted sand deserts, stretches across the southwest of Oman.

The climate in Oman varies considerably, from arid desert conditions in the lowlands to more temperate climates in the higher elevations of its mountains. Generally, the coastal regions, including the capital city of Muscat, experience a hot desert climate with year-round elevated temperatures [32]. In contrast, the climate in the mountainous areas is milder and cooler, with a more temperate environment. Oman spans an area of 309,500 km², and its agricultural sector covers approximately 14,662 km² or 4.7% of the total land area. A vast area of cultivated land is permanent meadows and pastures (92.1%), followed by arable land (5.6%) and permanent crops (2.3%) (https://wsp.mafwr.gov.om/).

This study focused on wheat cultivation and used a random point extension in ArcGIS Pro 3.3 to model the bioclimatic factors and generate 110 points. The dataset was randomly divided into two, with 70% of the data for training the model and 30% for validation, as illustrated in Fig 1.

### 2.2. Data collection and spatial processing

This study used the wheat distribution and inventory data from the most recent Agricultural Census published by the Ministry of Agriculture, Fisheries Wealth and Water Resources of Oman (https://www.mafwr.gov.om/). The distributions of wheat is still grown on the same locations as checked by personal communication to the Ministry of Agriculture and Fisheries Wealth and Water Resources (2025) (S1 in S1 File). The wheat-growing season in Oman spans November to April, and the critical factors influencing wheat production are temperature, water availability (especially soil moisture), humidity, wind conditions and solar radiation. Because of Oman's predominantly arid climate, irrigation systems, not natural precipitation, provide a significant proportion of the water required for wheat cultivation.

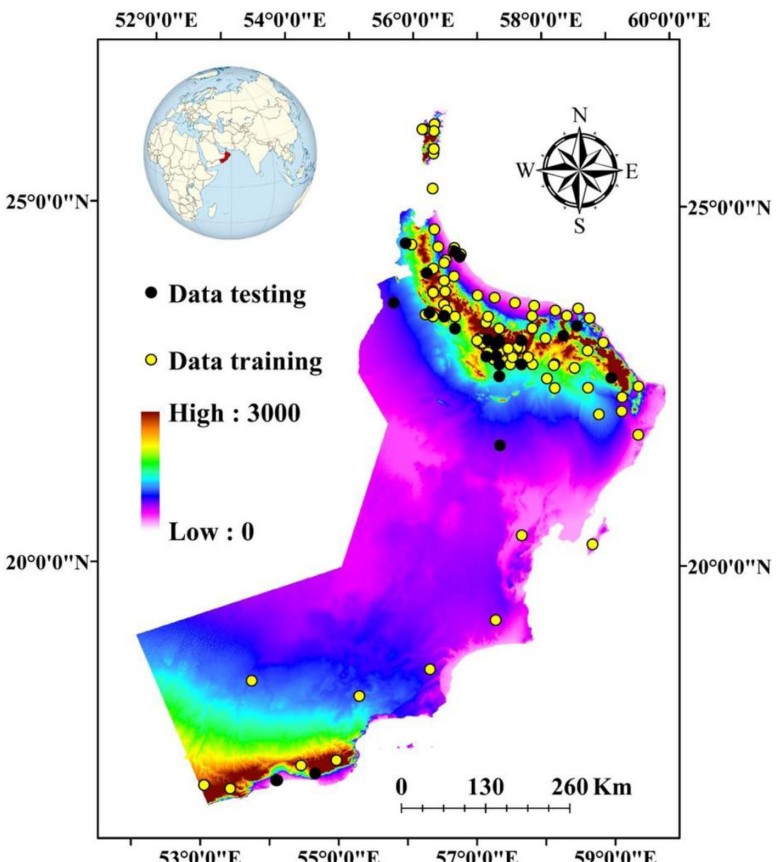

**Fig 1. Locations within the study area featuring the elevation map and wheat data used to train and test the model (Esri ArcGIS Pro 2024).**

This study assessed various bioclimatic factors, including annual mean temperature, temperature seasonality, minimum temperature in the coldest month, mean temperature for the coldest quarter, annual precipitation, precipitation during the wettest month and precipitation seasonality. The bioclimatic variables were from the WorldClim database (https://www.worldclim.org/), a global repository of high-resolution climate and meteorological data (S2, S3, S4 in S1 File and S2 File). The high spatial resolution of the WorldClim dataset makes it well-suited for spatial modelling and cartographic applications. In our study, we utilized the (CMIP6) framework for the climate changes models, which provides advanced simulations of climate conditions. Table 1 provides an overview of the critical bioclimatic variables essential for predicting the suitability of areas for wheat production. The nine bioclimatic factors integrated into the predictive models for wheat production and distribution in Oman played a vital role due to their profound effect on different physiological stages.

## 2.3. Multicollinearity analysis

The study performed a multicollinearity analysis using variance inflation factors (VIFs) to evaluate the influence of specific variables on the accuracy of the final wheat distribution maps. This technique helped identify the instances where multiple variables provide redundant information. Any variable with a VIF exceeding the threshold of 7.5 was excluded from further analysis to avoid multicollinearity [33–35]. The first step in the machine learning process was performing the VIF analysis to assess the inter-correlation among the nine bioclimatic variables selected based on their relevance to the theoretical framework (Table 1).

**Table 1. The bioclimatic factors crucial for wheat production in Oman.**

| Factor | Definition |
|---|---|
| BIO1 | Annual Mean Temperature |
| BIO4 | Temperature Seasonality (standard deviation ×100) |
| BIO6 | Minimum Temperature in the Coldest Month |
| BIO11 | Mean Temperature for the Coldest Quarter |
| BIO12 | Annual Precipitation |
| BIO13 | Precipitation in the Wettest Month |
| BIO15 | Precipitation Seasonality (Coefficient of Variation) |
| BIO16 | Precipitation in the Wettest Quarter |
| BIO19 | Precipitation in the Coldest Quarter |

## 2.4. Maximum entropy method

This study used the MaxEnt model to predict the suitability of wheat cultivation in Oman across four periods: 1970–2020, 2021–2040, 2041–2060 and 2061–2080. This approach incorporated historical and projected bioclimatic data to evaluate how environmental changes might influence wheat distribution. The MaxEnt method is a widely used machine-learning technique to model species distribution and is suitable for predicting wheat suitability under varying environmental scenarios [36–38]. The underlying principle of MaxEnt is the maximum entropy theory, which posits that when estimating a probability distribution, the distribution should be chosen to maximise entropy, subject to the given constraints [39–42]. This approach ensures that the model's conclusions are based solely on the data without introducing any unwarranted assumptions or biases.

## 2.5 Evaluating the MaxEnt model

The MaxEnt model is a probabilistic approach used extensively in species distribution modelling (SDM) and ecological and environmental studies, such as studies on the impact of climate change on food production, including wheat. The framework is grounded in information theory and statistical mechanics and generates predictions with minimal assumptions while accurately representing the observed data.

The area under the curve (AUC) was used to evaluate the effectiveness of the MaxEnt model by measuring its ability to distinguish between species presence and background points based on environmental suitability [43,44]. The assessment involved generating a receiver operating characteristic curve illustrating the relationship between the true and the false positive rates across various probability thresholds. The AUC gave a single value summarising the model performance, ranging from 0.5 (random prediction) to 1 (perfect prediction) [45,46]. Models with an AUC above 0.7 are considered accurate. For instance, an AUC of 0.85 indicates an 85% likelihood of correctly ranking presence points above background points. This metric offers a reliable numerical assessment of the accuracy of the MaxEnt model in predicting species distribution using environmental data [47]. To categorize wheat suitability into five distinct levels—extremely high, high, medium, low, and extremely low—we utilized the Natural Breaks (Jenks) classification method in ArcGIS Pro 3.3. This technique optimizes the classification by minimizing variance within each class while maximizing variance between classes, thus revealing natural groupings within the data. The Jenks algorithm determined the thresholds for each suitability level, effectively dividing the range of suitability scores into meaningful categories based on the data's inherent distribution. This method ensures that the classification accurately reflects the underlying spatial patterns of suitability, maintaining clarity in interpreting wheat cultivation potential across the study area. Fig 2 summarises the methodology used to evaluate wheat cultivation suitability in Oman by incorporating historical and projected bioclimatic factors in a step-by-step analysis.

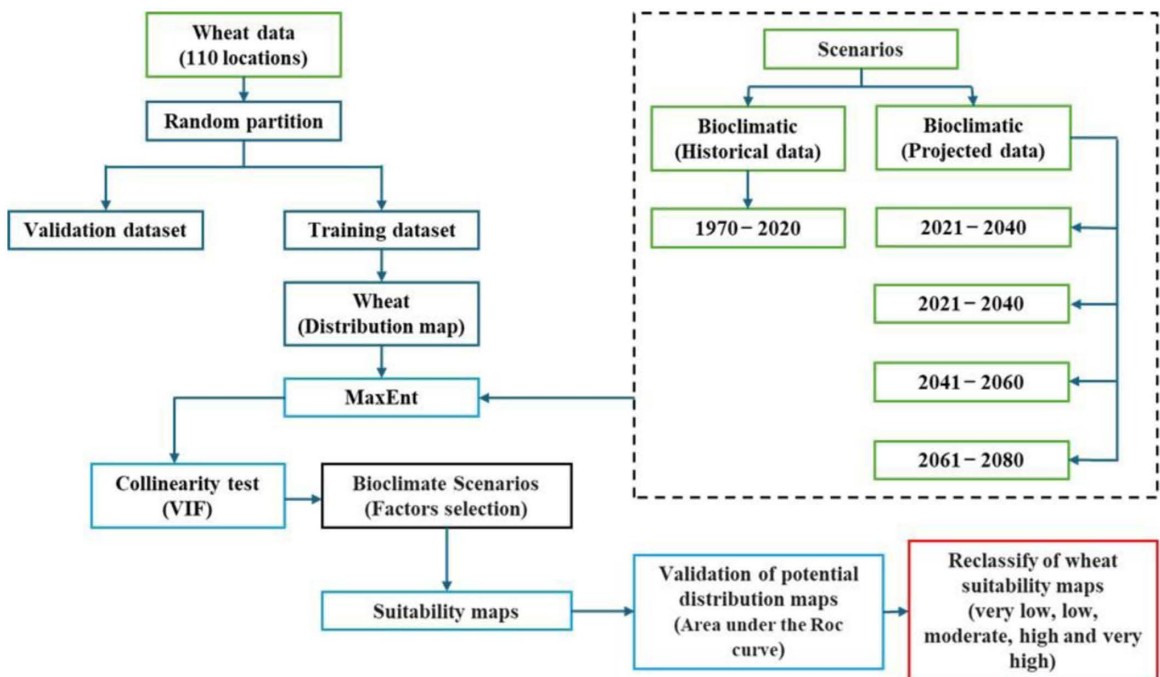

**Fig 2. Schematic representation of the sequential steps in the analytical process.**

## 3. Results and discussion

### 3.1. Multicollinearity analysis

The multicollinearity analysis used variance inflation factors (VIFs) to identify variables providing redundant information that might affect the accuracy of the final wheat distribution maps [48]. Variables exceeding the 7.5 threshold were excluded from further analysis [49]. Following this assessment, five of the nine variables were deemed suitable for inclusion in the MaxEnt model and successfully passed the VIF test. The selected variables were temperature seasonality (SD × 100), mean temperature for the coldest quarter, precipitation in the wettest month, precipitation seasonality, and precipitation in the coldest quarter. The assessment indicated that these factors are the most suitable for wheat cultivation under changing climate conditions. Including these variables enabled the model to accurately predict the geographical distribution of wheat (Table 2).

### 3.2. MaxEnt method

**3.2.1. Model evaluation.** The MaxEnt model effectively predicted wheat distribution in the study region, with strong outcomes across all periods. The model achieved an AUC score of 0.82 for the reference period (1970–2020), indicating high precision in identifying regions suitable for wheat cultivation (Figure and demonstrating the model's success in capturing wheat distribution patterns. Guo et al. [20] recognised the MaxEnt model as effective for forecasting wheat cultivation trends, especially regarding climate change adaptation and agricultural strategy development.

The AUC decreased slightly to 0.81 (Fig 3) for the 2021–2040 period, reflecting potential changes in environmental conditions that might affect wheat distribution patterns. Despite this, the model's predictive ability remained robust. For the 2041–2060 period, the AUC was again 0.82, maintaining a robust predictive accuracy despite the projected climate change impacts [50].

**Table 2. The variables selected for the model predicting the geographical distribution of wheat.**

| Factor | VIF |
|---|---|
| Annual Mean Temperature | 8.7 |
| Temperature Seasonality (SD × 100) | 2.61 |
| Minimum Temperature in the Coldest Month | 9.1 |
| Mean Temperature for the Coldest Quarter | 4.10 |
| Annual Precipitation | 10.2 |
| Precipitation in the Wettest Month | 5.05 |
| Precipitation Seasonality | 1.55 |
| Precipitation in the Wettest Quarter | 12.01 |
| Precipitation in the Coldest Quarter | 3.06 |

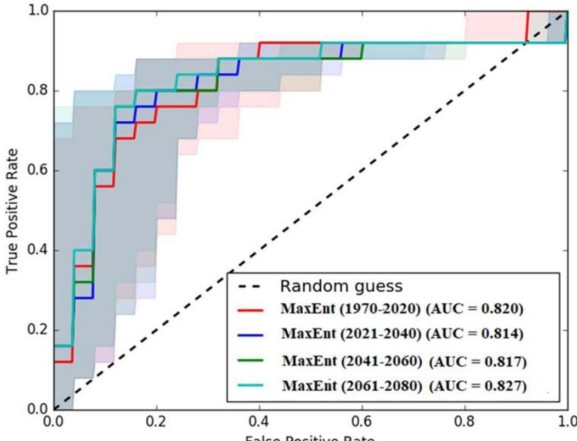

**Fig 3. Model assessment showing the AUC outcomes based on historical and projected bioclimatic variables (Esri ArcGIS Pro 2024).**

The AUCs remained consistent throughout these periods, suggesting that the bioclimatic variables of the model effectively predicted the wheat-growing potential. After 2061, the AUC increased slightly to 0.83, potentially indicating that the wheat distribution patterns are more distinct in the projected climate scenarios or that the model's environmental factors gain greater importance as climate change progresses. These results underscore the effectiveness of the MaxEnt model in predicting wheat distribution amidst climate change, providing valuable insights into long-term agricultural planning and sustainable practices. Generally, MaxEnt is one of the suitable tools for modelling the water-energy-food nexus for wheat production in arid and semiarid environments [51].

**3.2.2. Temporal analysis of wheat cultivation suitability areas.** The temporal assessment of suitability zones by the MaxEnt model (1970–2080) revealed distinct patterns in the percentage and spatial extent (km²) across five suitability categories: extremely high, high, moderate, low and extremely low. Fig 4 shows the wheat-growing suitability maps generated using the MaxEnt method. The changes offer valuable insights into the evolution of agricultural potential, with significant implications for land-use planning, agricultural policy development [10] and wheat production under climate change scenarios [52,53].

Regions with optimal environmental conditions for wheat growth are classified as areas with very high suitability. Between 1970 and 2020, only three per cent of the total area, approximately 9,285 km², was deemed highly suitable for wheat farming. However, this classification increased slightly to four per cent between 2041–2060, expanding the area to

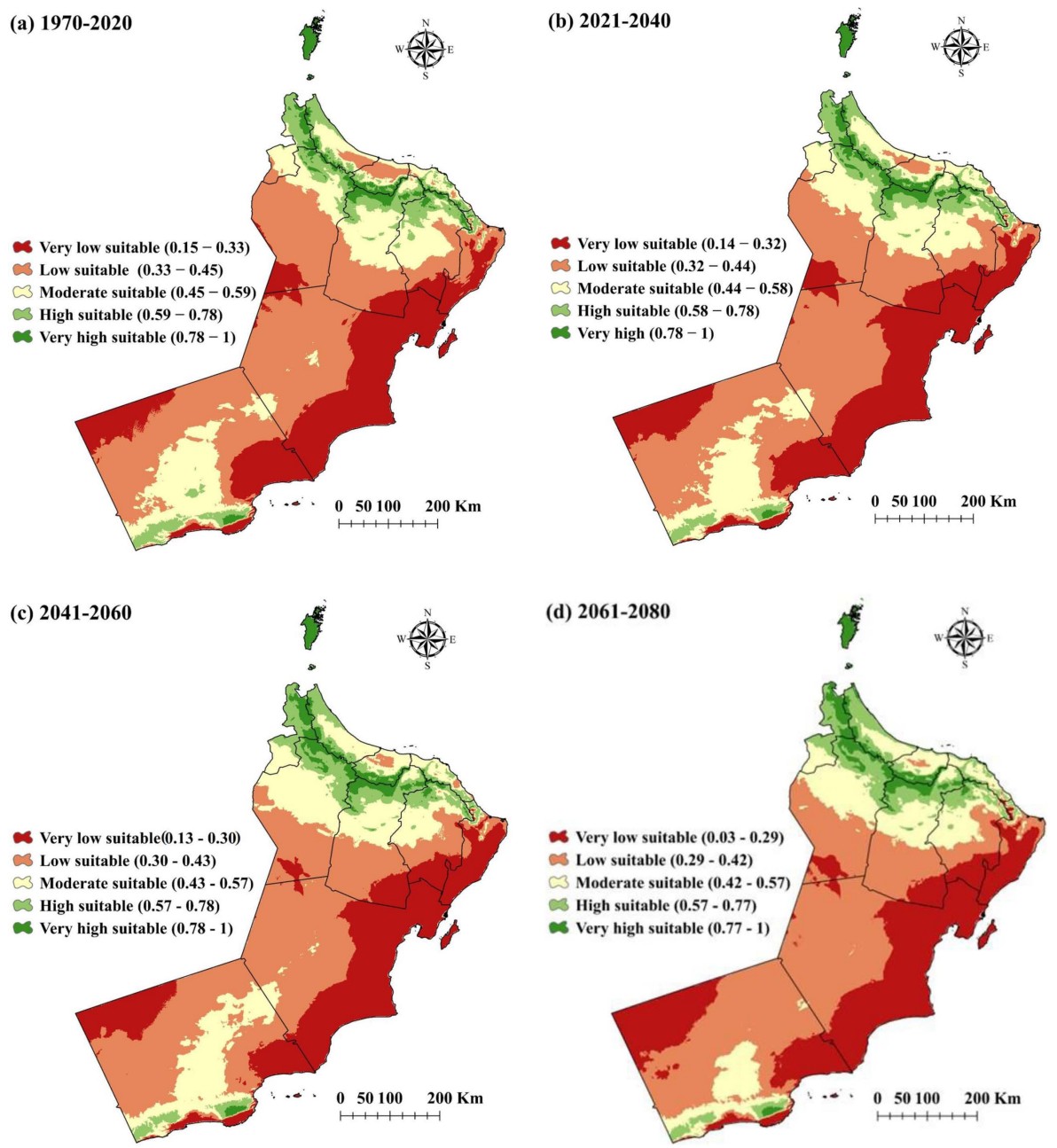

**Fig 4. MaxEnt-generated wheat suitability maps (Esri ArcGIS Pro 2024) with classification predicted for different period of time (the wheat distribution is given between brackets).** (a) from 1970- 2020, (b) from 2021-2040, (c) from 2041-2060, (d) from 2061-2080.

12,380 km². This suitability remained stable throughout 2061–2080 (Figs 5a and 5b). The modest increase suggests that specific environmental factors, such as rainfall patterns, soil quality or temperature variations, might have improved and enhanced the land suitability for wheat cultivation.

Despite this expansion, the total area of very high suitability remains small compared to other suitability categories, indicating that only a limited fraction of land is highly suitable for wheat farming. These optimal zones should be prioritised

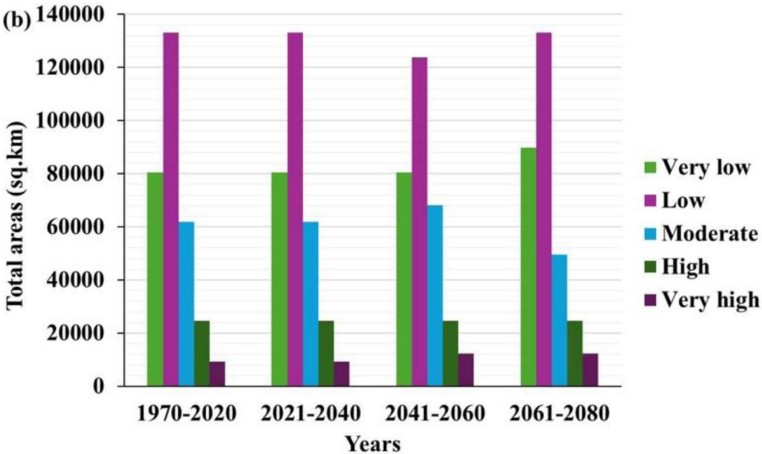

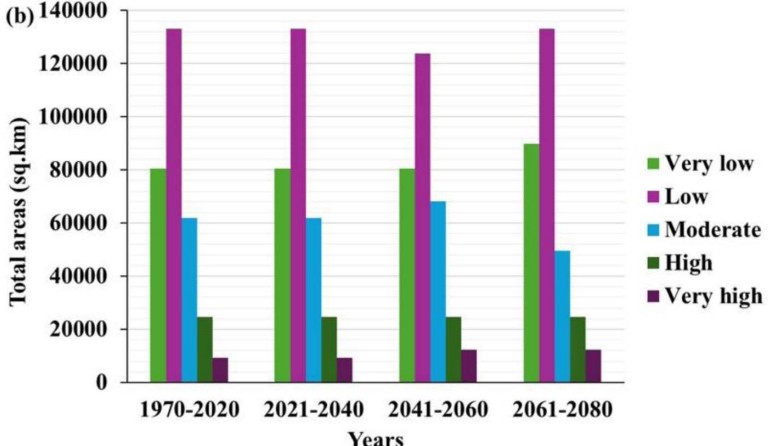

**Fig 5.** a. The percentage of temporal wheat cultivation suitability showing the land distribution across five categories, very high, high, moderate, low and very low suitability, from 1970 to 2080 based on the MaxEnt model projections. b. Temporal wheat cultivation suitability by area (km2). The chart categorises the land area (in km²) into five suitability categories, very high, high, moderate, low and very low, for wheat cultivation from 1970 to 2080 based on the MaxEnt model projections.

for the implementation of innovative farming techniques, including precision agriculture, to maximise yields and ensure sustainable wheat production [53,54]. The areas classified as highly suitable for wheat cultivation remained constant across all periods, covering eight per cent or 24,760 km² of the total land area. This stability suggests that the environmental conditions necessary for moderately optimal wheat growth were resilient to climate or ecological changes (Figs 4a and 4b). The regions maintaining their extent without fail qualify as highly suitable for wheat cultivation, although the yield might be lower than the very high suitability areas. Prudent management of water and soil resources is crucial to ensure long-term wheat farming productivity in these areas.

The intermediate suitability zones accounting for 20% of the total area (about 61,900 km²) from 1970 to 2020 displayed considerable variability. This percentage remained stable until 2040 and increased slightly to 22% (about 68,090 km²) during 2041–2060 (Figs 4a and 4b). The projections indicated a 16% decline (approximately 49,520 km²) in 2061–2080. The fluctuation highlights the vulnerability of these regions towards changes in climate and environmental conditions. The temporary increase in moderately suitable areas during 2041–2060 might be due to favourable weather patterns or

better land management practices. However, the subsequent decline suggests growing challenges, possibly due to rising temperatures, soil degradation or less water availability. The lower percentage in moderately suitable areas emphasises that wheat production would be increasingly challenging without significant interventions, such as using drought-resistant wheat cultivars, advanced irrigation systems and adopting climate-smart agriculture approaches and technologies [55–58].

Areas with low suitability consistently dominated the landscape, accounting for 40%–43% of the total area throughout the study period, corresponding to 133,085 km² during the 1970–2020 and 2021–2040 periods. Between 2041 and 2060, this area decreased marginally to 123,800 km² before returning to 133,085 km² in 2061–2080 (Figs 4a and 4b). The persistent classification indicates that a significant portion of the land was only marginally suitable for wheat production, possibly due to poor soil conditions, limited water resources or unsuitable climate. The slight decrease in low-suitability areas in the mid-century could be due to temporary improvements in certain environmental conditions. However, these enhancements are insufficient for elevating large areas to higher suitability classifications. Significant investments in technologies such as advanced irrigation systems [59] or soil improvement methods are necessary to make future wheat cultivation feasible in low-suitability areas [60]. Another alternative is implementing machine learning algorithms for sustainable and economical wheat cultivation [59]

A comparison of percentage and area-based statistics revealed the extent of changes in land suitability and their geographical implications for wheat production. Although the proportion of areas classified as very high and high suitability remained relatively constant or increased slightly, their land coverage remained limited, indicating that only an insignificant fraction of the total area is highly productive (Figs 4a and 4b). It is noteworthy that the maximum area for wheat production is estimated to be approximately 2700–3000 km2 under the current average wheat production and average consumption of wheat in Oman, which is only approximately 10–15% of very high suitability. It is crucial to prioritise the regions for wheat cultivation to optimise yields and ensure food security. The fluctuations in moderately suitable areas and the continued dominance of low and very low suitability zones highlight the challenges that climate change and environmental degradation pose to agricultural practices [61]. The contraction of moderately suitable regions and the expansion of areas with very low suitability emphasise the need for adaptable management strategies. These strategies should include farming methods resilient to climate change, improving soil quality and efficient water management infrastructure to sustain agricultural output amidst evolving climate conditions [62].

Furthermore, the marginal increase in very low suitability areas towards the end of the century suggests the necessity for long-term planning and policies that address the broader impacts of climate change, which may encompass diversifying crops, changing land-use patterns and implementing innovative land management practices to build resilience in low-suitability regions [10,63].

### 3.5. Wheat versus bioclimate factors (1970–2020)

**3.5.1. Wheat probability versus temperature seasonality.** The AUC in Fig 6a illustrates the inverse correlation between wheat distribution and temperature seasonality (SD × 100). Regions with consistent temperatures were more conducive to wheat growth, as demonstrated by the high occurrence probability (exceeding 0.99%) in areas where temperature seasonality was below 200°C. However, this probability declined as seasonality exceeded 300°C, with the curve plateauing at higher values. The likelihood of wheat presence declined sharply above 400°C (SD × 100), indicating that increased temperature fluctuations reduced the suitability for wheat cultivation [64] and highlighting wheat's vulnerability to unstable climatic conditions (Fig 6a). Unstability of temperature and Elevated temperatures during the anthesis or grain filling stage reduced the photosynthetic rate by compromising thylakoid membrane integrity. Yield-related traits, including seed set percentage and individual grain weight, were reduced due to high temperatures during the anthesis or grain filling stage, respectively. Similar conclusions were made by Gupta et al. [65] that higher temperatures lead to increased transpiration, which can trigger drought and subsequently

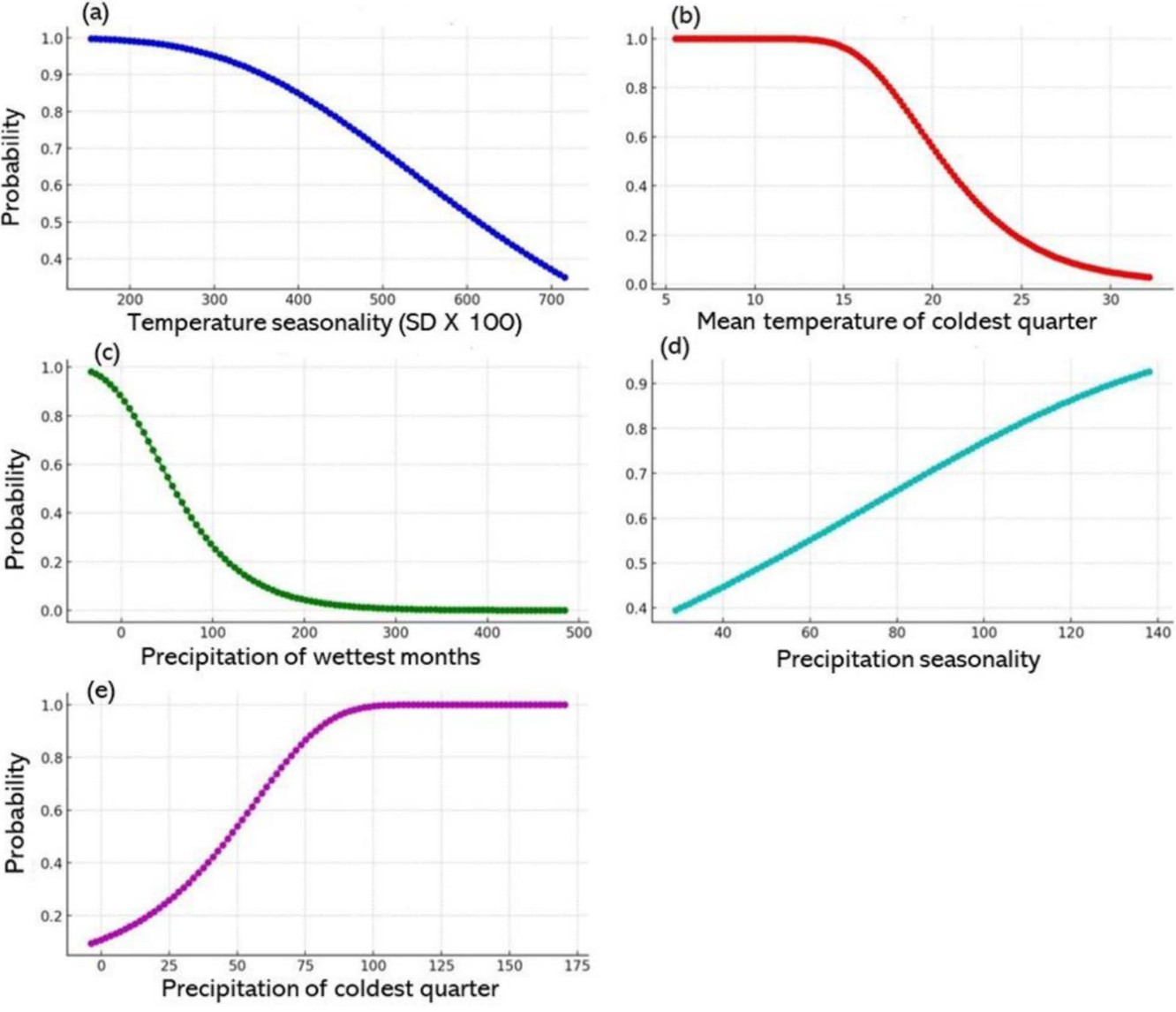

**Fig 6. The association between wheat distribution and bioclimatic factors for 1970–2020.** Panels a–e shows the correlations between wheat distribution and various climatic variables: temperature seasonality, mean temperature for the coldest quarter, precipitation in the wettest month, precipitation seasonality, and precipitation in the coldest quarter, respectively.

reduce production of wheat. In addition, predictions indicate that in the next 20–30 years, a 2°C temperature shift might result in a significant water crisis.

 **3.5.2. Wheat probability versus the mean temperature for the coldest quarter.** The AUC in Fig 6b shows the relationship between wheat occurrence probability and the mean temperature for the coldest quarter. Optimal wheat cultivation occurred at lower temperatures of 5.5°C–10°C, with the probability approaching 100%. The probability gradually decreased as the temperature exceeded 10°C, with a sharper decline beyond 16°C, indicating that higher temperatures during the coldest quarter reduced wheat suitability. The trend in the AUC shows a gradual decrease in

probability with higher temperatures, particularly above 16°C. In summary, cooler temperatures favoured wheat cultivation, while higher temperatures reduced its feasibility.

**3.5.3  Wheat probability versus precipitation in the wettest month.**  The AUC in Fig 6c illustrates the relationship between the probability of wheat occurrence and the precipitation in the wettest month, where the highest rainfall was 101.70 mm. The AUC provides insights into the general likelihood of wheat presence across varying rainfall levels, highlighting the significant influence of precipitation on suitability. An elevated AUC suggests a strong correlation between the predictor variable (precipitation) and the outcome (wheat probability). In this case, the AUC shows that the higher rainfalls during the wettest month significantly influenced the probability of wheat occurrence, with a gradual decline in wheat likelihood with higher precipitation levels. This pattern suggests that moderate rainfalls were more conducive to wheat occurrence, while extreme values, either too little or too much, reduced suitability (Fig 6c).

**3.5.4.  Wheat probability versus precipitation seasonality (coefficient of variation).**  The correlation between wheat probability and precipitation seasonality, quantified by the coefficient of variation, gave an AUC of 0.73%. The AUC metric also suggests a link between a marked variability in rainfall, indicated by a higher coefficient of variation, and a higher probability of wheat occurrence. This finding shows that environments with marked seasonal precipitation changes might provide favourable conditions, possibly by ensuring water availability throughout the year. The upward trend of the probability curve supports the notion that increased rainfall variability could enhance the chances of wheat survival, possibly by maintaining consistent moisture levels during critical growth phases.

**3.5.5.  Wheat probability versus precipitation in the coldest quarter.**  The relationship between the probability of wheat occurrence and precipitation during the coldest quarter (Fig 6e) revealed a strong positive correlation. At low rainfall levels, the likelihood of wheat occurrence was less than 0.20%, increasing to 0.40% at 39.25 mm and 0.61% at 55.4 mm. The probability increased to over 0.90% when precipitation surpassed 80.51 mm, indicating highly favourable conditions. The probability approached 1.0 when the rainfall exceeded 100 mm, suggesting a near-certain wheat occurrence. The AUC of 126.10 reinforces this strong correlation, highlighting that the higher precipitation during this period significantly enhances the likelihood of wheat presence.

## 3.6.  Wheat versus bioclimate factors (2021–2040)

**3.6.1.  Wheat probability versus temperature seasonality (SD × 100).**  The AUC analysis in Fig 7a illustrates the relationship between temperature seasonality and wheat suitability for 2021–2040. The significant seasonal temperature variations in this timeframe indicated by higher temperature seasonality adversely affect wheat growth [66]. Areas with minimal seasonal temperature fluctuations exhibited high wheat suitability, reaching 0.95%, suggesting that wheat thrives in climates with stable temperatures throughout the year. However, wheat suitability declined sharply as seasonality values exceeded 300, dropping to 0.44% at seasonality of 700 mm. The critical threshold occurred between 300 and 350 mm, and the suitability remained above 0.7%. Beyond this point, suitability decreased rapidly, highlighting that marked temperature fluctuations substantially limited wheat's adaptability (Fig 7a). The identified threshold for 2021–2040 offers valuable insights for agricultural planning and emphasises that regions with lower seasonality were more favourable for sustainable wheat cultivation.

**3.6.2.  Wheat probability versus the mean temperature for the coldest quarter.**  The probability for wheat suitability remained the highest at 1.0 until the mean temperature for the coldest quarter reached 12°C (Fig 7b). Beyond this threshold, suitability decreased significantly after 17°C and reached its lowest point when the temperature exceeded 25°C. This pattern suggests that wheat cultivation was best suited to regions where the coldest quarter temperatures were below 12°C. Once the temperature exceeds this range, particularly above 17°C–18°C, the suitability for wheat cultivation decreased markedly, indicating that warmer regions were less conducive to wheat production (Fig 7b).

**3.6.3.  Wheat probability versus precipitation in the wettest month.**  Fig 7c shows a clear inverse relationship between wheat suitability and rainfall during the wettest month. The probability of wheat suitability peaked at 0.85% when

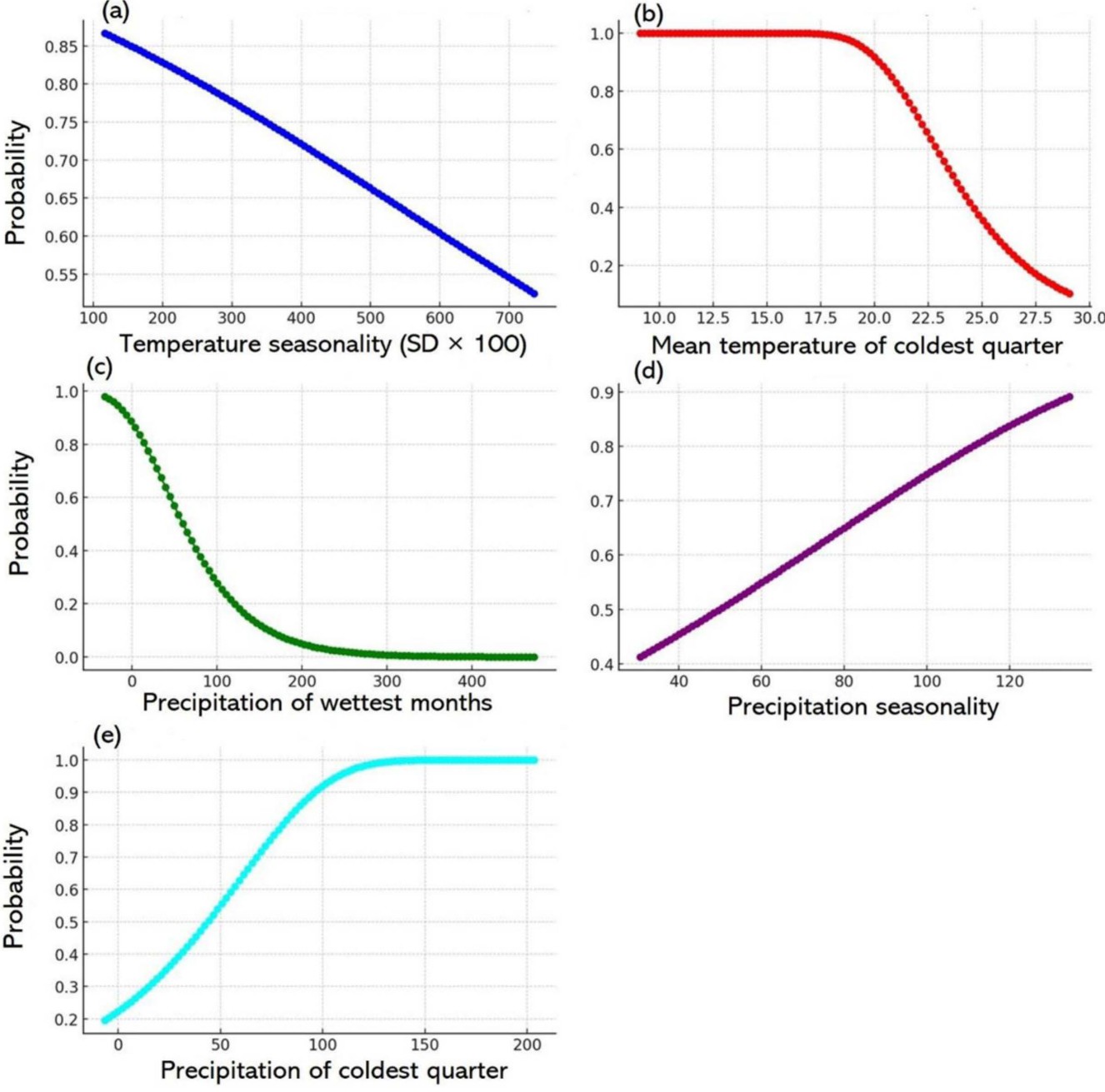

**Fig 7. Wheat distribution and bioclimatic factors for 2021–2040.** Panels a–e show the associations between wheat distribution and key climatic parameters: temperature seasonality, average temperature for the coldest quarter, precipitation in the wettest month, precipitation seasonality, and precipitation in the coldest quarter, respectively.

rainfall was 3.6 mm. However, the likelihood of suitability gradually diminished with higher precipitation. The probability decreased to 0.73% at 24 mm of rainfall and continued to drop to 0.62% at 39 mm. This negative trend continued with more substantial rainfalls, where the probability decreased to 0.39% at 75 mm and 0.26% at 100 mm. As the precipitation exceeded 150 mm, wheat suitability declined significantly, with the likelihood approaching zero when rainfall exceeded

250 mm. These findings underscore that wheat cultivation was most viable in areas with moderate rainfalls during the wettest month, while higher precipitation levels reduced the likelihood of successful wheat farming [67].

**3.6.4. Wheat probability versus precipitation seasonality.** The relationship between wheat suitability and precipitation seasonality, indicated by the coefficient of variation, revealed that regions with higher rainfall variability provided suitable conditions for wheat cultivation. The probability of wheat suitability remained relatively modest, approximately 0.44%, in areas with lower precipitation seasonality, where the coefficient of variation ranged from 28 to 40 (Fig 7d). However, the likelihood of wheat suitability increased with higher precipitation variability, reaching approximately 0.87% when the coefficient of variation was 136. This trend suggests that wheat might grow better in regions with more pronounced seasonal rainfall fluctuations, possibly because the conditions ensured water availability during critical growth stages, thus enhancing overall suitability for cultivation.

**3.6.5. Wheat probability versus precipitation in the coldest quarter.** An analysis of the correlation between wheat suitability and precipitation during the coldest quarter revealed that wheat thrived in areas with minimal rainfall (Fig 7e). The likelihood of wheat suitability remained high (over 0.8%) at low precipitation levels of 0–50 mm, suggesting that wheat flourished in drier conditions during the coldest quarter. However, the probability gradually decreased as rainfall exceeded 50 mm. The likelihood decreased to 0.5% at 100 mm rainfall, indicating that the higher moisture during the coldest quarter was less conducive to wheat growth. Excessive moisture levels could cause waterlogging, soil saturation and root diseases that hindered wheat development. The probability approached zero when precipitation exceeded 200 mm, suggesting that areas with substantial rainfall during the coldest quarter were unsuitable for wheat cultivation (Fig 7e).

**3.7. Wheat versus bioclimate factors (2041–2060)**

**3.7.1. Wheat probability versus temperature seasonality (SD×100).** The projection for wheat suitability in 2041–2060, shown in Fig 8a, indicates a decline in suitability as temperature rises [68]. Wheat is susceptible to temperature consistency, where the optimal wheat suitability was at a seasonality range of 116–400. Wheat suitability was below 0.7% outside this range, marking a threshold where significant temperature fluctuations began to hinder wheat growth. In regions with seasonality exceeding 600, wheat suitability drops below 0.6%, suggesting that extreme temperature variations might render some areas unsuitable for wheat cultivation by 2041–2060. The MaxEnt model, through AUC analysis, highlights the importance of temperature stability in future wheat production and underscores the possible need for adaptive strategies in areas vulnerable to high variability.

**3.7.2. Wheat probability versus mean temperature for the coldest quarter.** The relationship between wheat probability and the average temperature in the coldest quarter shows optimal conditions between 9°C and 13.75°C, with 100% probability (Fig 8b). Suitability declined as the temperature exceeded 13.75°C. Wheat still grew at 14°C–15°C, but with lower suitability. When the temperature exceeded 19°C, the probability fell below 0.95%, dropping to 0.60% at 22.4°C and under 0.30% above 26°C. Areas with cold quarter temperatures exceeding 28°C were highly unsuitable for wheat cultivation [69].

**3.7.3. Wheat probability versus precipitation in the wettest month.** Fig 8c shows a clear trend between precipitation in the wettest month and wheat suitability, with favourable conditions gradually declining with higher rainfalls. Optimal wheat suitability occurred with minimal or no precipitation, where 0 mm and 8 mm rainfall correspond to a probability higher than 0.80%. As precipitation increased, suitability declined from 0.77% at 18 mm to 0.57% at 49 mm. The decline was apparent when rainfall exceeded 50 mm, reaching 0.27% at 100 mm and decreasing further at higher precipitation levels. Rainfall above 200 mm results in extremely low probabilities, indicating unsuitable conditions for wheat cultivation. These results suggest that areas with lower rainfall in the wettest month were more conducive to wheat growth.

**3.7.4. Wheat probability versus precipitation seasonality.** The data in Fig 8d shows a clear positive correlation between precipitation seasonality and wheat suitability. The likelihood of wheat suitability increased with pronounced

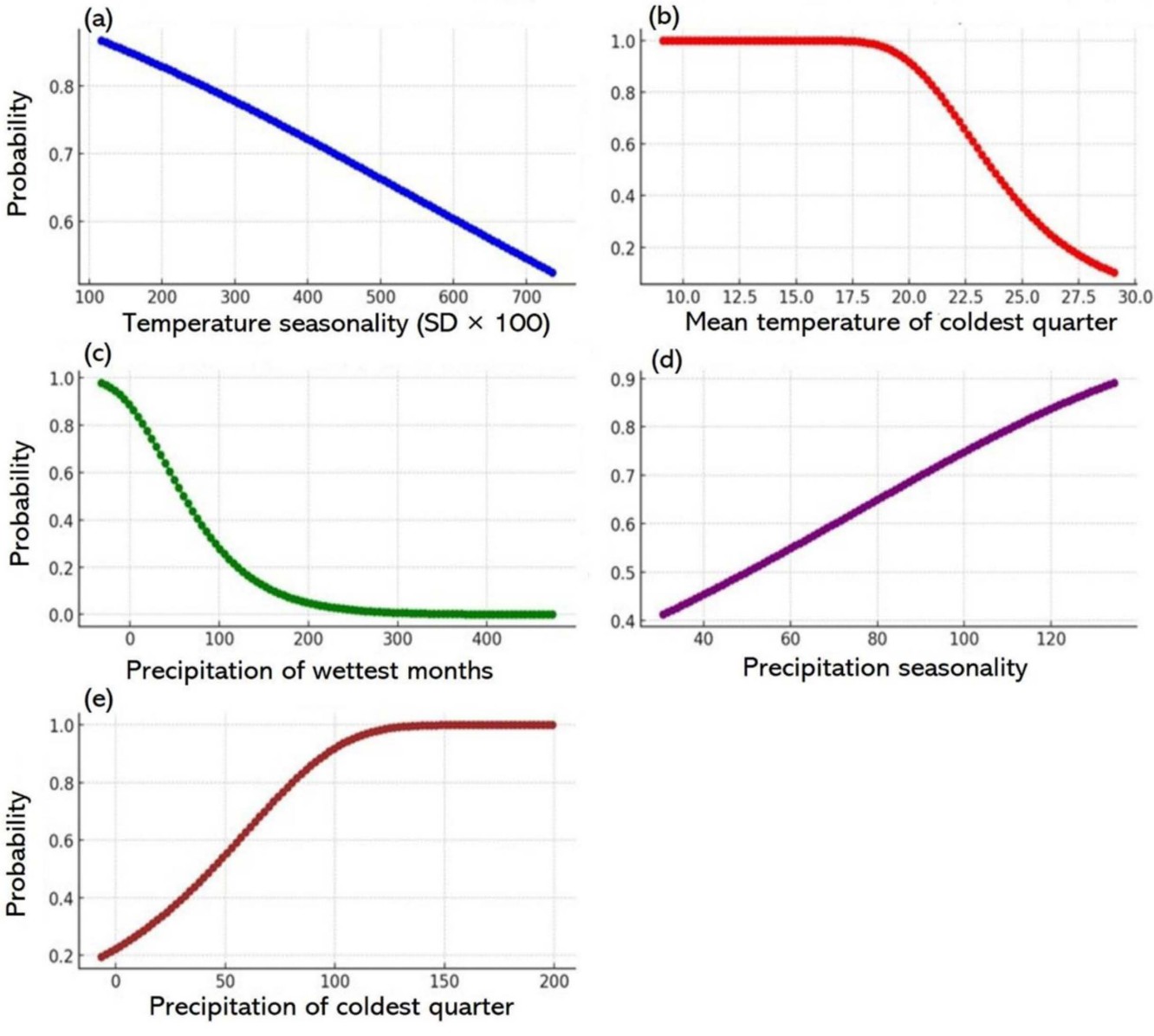

**Fig 8. Wheat distribution and bioclimatic factors for 2041–2060.** Panels a–e shows the associations between wheat distribution and key climatic parameters: temperature seasonality, average temperature for the coldest quarter, precipitation during the wettest month, precipitation seasonality, and precipitation in the coldest quarter, respectively.

seasonal rainfall variability, suggesting that areas with greater precipitation fluctuations were more conducive to wheat growth. In regions with moderate precipitation seasonality (30–60 mm), the probability gradually rose from 0.41% to 0.55%. When seasonality exceeded 60 mm, the probability increased dramatically, reaching 0.65% at 80 mm and 0.70% at 90 mm. These findings indicate that areas with higher seasonal precipitation fluctuations were more suitable for wheat cultivation.

The wheat probability continued to rise with higher levels of precipitation seasonality. For instance, when seasonality reached 100 mm, the probability was 0.79% and continued to increase as seasonality rose, reaching 0.85% beyond

120 mm and peaking at 0.89% when seasonality values were around 134 mm (Fig 8d), indicating that areas with the most pronounced precipitation seasonality were expected to be the most conducive to the modelled scenario. Generally, there was an association between increased precipitation seasonality and a higher probability of suitability. Areas with more significant seasonal rainfall fluctuations became increasingly favourable for wheat growth, with the highest probabilities observed at the upper extremes of the seasonality range, indicating that the modelled conditions were most suited to environments with evenly distributed rainfall throughout the year.

### 3.7.5. Wheat probability versus precipitation in the coldest quarter. Fig 8e shows a distinct positive association between precipitation during the coldest quarter and wheat probability. The wheat probability began at 0.20% in areas with very low rainfall, such as 6 mm, indicating that locations with minimal precipitation during the coldest quarter were less conducive to the modelled conditions. A marginally higher rainfall resulted in gradually higher probability, although it remained under 0.3% until rainfall of approximately 15 mm, suggesting that areas with low rainfalls remained relatively unfavourable for wheat growth.

As precipitation levels increased from 15 mm to 50 mm, the probability improved significantly, with values rising from 0.32% to 0.55%, indicating that moderate rainfall amounts correspond to a higher likelihood of favourable conditions and a trend towards better suitability. This precipitation range appears more advantageous for the modelled scenario, as the probability surpassed the 0.5% threshold (Fig 8e). The probability gradually increased as rainfall exceeded 50 mm. For instance, the probability was 0.62% at 60 mm and 0.80% at 80 mm rainfall, suggesting that higher precipitation during the coldest quarter enhanced wheat suitability and a significant positive correlation between rainfall and habitability within this range. The probability consistently exceeded 0.90% when precipitation exceeded 100 mm. For instance, the probability was 0.92% at 99 mm rainfall and continued to rise rapidly thereafter. As precipitation reached and surpassed 150 mm, the probability approached 100%, signalling a near-ideal suitability.

## 3.8. Wheat versus bioclimate factors (2061–2080)

### 3.8.1. Wheat probability versus temperature seasonality (SD × 100). The analysis of temperature seasonality and wheat suitability for 2061–2080 revealed an inverse relationship (Fig 9a). The likelihood of suitable conditions for wheat cultivation diminished gradually with higher temperatures. The highest probability of wheat suitability (0.96%) in areas with lower temperature variability (approximately 108) suggests optimal growing conditions. This high probability remained relatively stable for temperature seasonality values up to approximately 250, with the likelihood generally exceeding 0.88%.

The likelihood of wheat suitability decreased noticeably when temperature seasonality exceeded 250 (SD × 100). The probability remained relatively high at 0.86% at a temperature seasonality of 300 but declined thereafter. Once temperature seasonality reaches 400, the probability drops to 0.75%, indicating that more extreme temperature fluctuations adversely impact wheat cultivation (Fig 9a). Much lower wheat suitability of 0.72% occurred as temperature seasonality exceeded 450, and the suitability diminished to 0.66% when temperature seasonality reached 500. These figures suggest that areas experiencing significant temperature fluctuations were progressively less suitable for wheat production. The most noticeable decline occurred when seasonality exceeded 600, with the probability plummeting to 0.56% and decreasing as seasonality intensified. As seasonality reached 750°C, the suitability decreased to 0.45%, indicating that regions with extreme temperature variations would face considerable challenges sustaining wheat crops.

The research findings suggest that wheat grows best in areas with stable temperatures where successful cultivation is highly probable. As temperature fluctuations became more pronounced, notably when seasonality surpassed 500 (SD × 100), the suitability for wheat farming decreased significantly. This trend implies that wheat production remained more favourable in regions with consistent thermal conditions, while areas with considerable temperature variability would have problems sustaining wheat crops in 2061–2080 (Fig 9a).

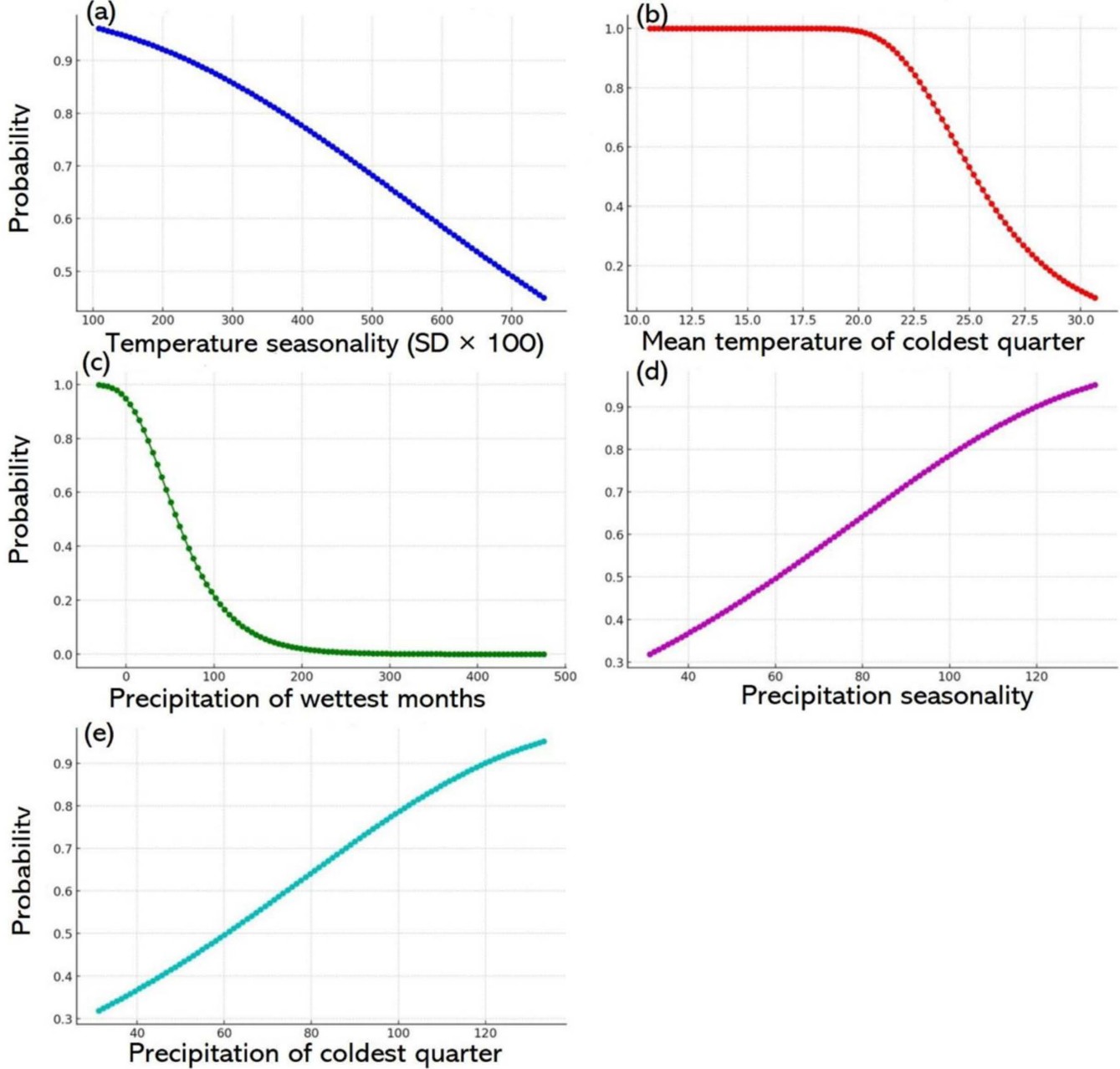

**Fig 9. Wheat distribution and bioclimatic factors for 2061–2080.** Panels a–e shows the associations between wheat distribution and key climatic parameters: temperature seasonality, average temperature for the coldest quarter, precipitation during the wettest month, precipitation seasonality, and precipitation in the coldest quarter, respectively.

**3.8.2. Wheat probability versus mean temperature of the coldest quarter.** The analysis revealed a strong correlation between the mean temperature for the coldest quarter and the likelihood of wheat suitability. The optimal conditions for wheat growth (100% probability of suitability) occurred when the average temperature range was

10.6°C–15.8°C (Fig 9b). The cooler temperatures during the coldest quarter provided ideal conditions for wheat cultivation, making these areas highly suitable for wheat growth.

The likelihood of wheat suitability decreased slightly as the average temperature exceeded 16°C. For instance, wheat suitability drops marginally to 0.99 even though it is still high at 17°C. Although the initial decline was modest, it indicates a gradually decreasing wheat suitability with higher temperatures despite the still favourable conditions for wheat growth (Fig 9b). The decline was more pronounced as the average temperature approached 20°C. At 20.72°C, the probability decreased to 0.97% and continued to decline with higher temperatures. At 21.33°C, the wheat probability dropped to 0.94%, demonstrating consistently lower wheat suitability as temperatures during the coldest quarter exceeded 20°C. Despite the relatively high suitability, the observed pattern indicates that elevated temperatures in the cold quarter adversely affect wheat growth (Fig 9b).

The suitability for wheat cultivation declined sharply when the temperature exceeded 22°C to 0.84% at 22.54°C and 0.72% at 23.56°C, a significantly lower wheat growth potential. These figures show that areas with cold quarter temperatures higher than 22°C are increasingly unsuitable for wheat cultivation. The most notable decline occurred at temperatures above 25°C, where the suitability dropped to 0.43% at 25.78°C and 0.32% at 26.8°C. Areas with cold quarter mean temperatures exceeding 28°C had extremely low probabilities of less than 0.2%, making these regions highly unfavourable for wheat production. Therefore, in 2061–2080, wheat cultivation would be most viable in cooler areas with mean cold quarter temperatures between 10.6°C and 15.8°C. Wheat suitability gradually declined as temperatures rose above 16°C, with more pronounced reductions after 20°C. Regions with cold-quarter temperatures above 25°C were markedly less suitable for wheat cultivation, with an almost zero probability of supporting wheat growth (Fig 9b). This analysis highlights the significant influence of cold-quarter temperatures on future wheat cultivation potential.

**3.8.3. Wheat probability versus precipitation during the wettest month.** There is a clear pattern of the association between precipitation in the wettest month and wheat suitability. The probability of suitable conditions for wheat cultivation gradually decreased with higher precipitation. For example, the probability dropped to 0.83% at 20 mm and 0.70% at 35 mm rainfall, suggesting that although areas with moderate rainfall retained some suitability for wheat growth, the likelihood of suitability steadily diminished as precipitation exceeded 20 mm (Fig 9c).

A more noticeable decline occurred when precipitation exceeded 50 mm. The lower probability of 0.56% at 50.7 mm and 0.43% at 66 mm rainfall indicates that areas with higher precipitation during the wettest month were increasingly less suitable for wheat cultivation, with conditions turning less favourable with higher rainfalls.

The reduced suitability is evident when precipitation surpasses 80 mm. Wheat suitability dropped to 0.32% at 81 mm of rainfall and 0.23% at 96.7 mm. The dramatically lower suitability suggests that regions with heavy rainfalls, particularly above 80 mm, were much less suitable for wheat cultivation. The probability for wheat cultivation approached zero in areas with extreme precipitation exceeding 150 mm. For instance, the probability was 0.07% at 150 mm of precipitation and continues to fall with higher rainfalls. The likelihood for wheat suitability was almost zero at 250 mm of rainfall, indicating that regions with excessive rainfalls during the wettest month were unsuitable for wheat production (Fig 9c).

**3.8.4. Wheat probability versus precipitation seasonality.** Fig 9d shows a robust and statistically significant positive correlation between wheat probability and precipitation seasonality (CV), suggesting a higher likelihood of wheat occurrence with higher precipitation variability. The 0.32 initial probability of wheat presence occurred at a low precipitation seasonality of approximately 31.17 CV. The rise in CV increased the wheat probability, crossing the 0.5 threshold at about 61.12 CV. This 0.5 threshold is a crucial juncture where precipitation fluctuations significantly influence the probability metric. Wheat probability continued to increase after surpassing the threshold value, reaching 0.75 at 95.18 CV and exceeding 0.95 at the maximum recorded precipitation seasonality of 133.38 CV (Fig 9d). The persistent upward trajectory indicates a robust association between higher precipitation variability and higher probabilities, potentially signifying favourable conditions for specific ecological, hydrological or climatic mechanisms. The relationship's lack of apparent saturation or plateau underscores the pivotal influence of precipitation seasonality on the wheat probability distribution.

It suggests that regions experiencing more pronounced seasonal precipitation variations were more prone to conditions supporting some environmental dynamics, such as ecosystem resilience, water resource availability or climate-related phenomena.

The simple and consistent correlation between wheat suitability and precipitation seasonality emphasised the significance of precipitation seasonality in influencing the probability of wheat distribution. This finding has significant ramifications in comprehending and forecasting climatic and environmental dynamics within the area.

**3.8.5. Wheat probability versus precipitation in the coldest quarter.** The correlation between rainfall during the coldest quarter and the likelihood of wheat suitability shows a clear positive trend with higher precipitation levels. The probability of wheat suitability is relatively modest in regions with lower rainfalls (31–40 mm), fluctuating between 0.32% and 0.37%. This observation suggests that although areas experiencing minimal precipitation during the coldest quarter provided moderate potential for wheat cultivation, wheat suitability improved with higher rainfalls (Fig 9e).

The probability of wheat suitability gradually increased as precipitation rose from 40 mm to 70 mm. For example, the likelihood increased from 0.40% at 45 mm rainfall to 0.57% at 70 mm. This pattern indicates that regions with higher precipitation during the coldest quarter were increasingly more suitable for wheat farming, surpassing the critical 0.50% threshold and signalling much better conditions for wheat growth.

Rainfalls between 70 mm and 100 mm significantly increased the likelihood of wheat suitability. The probability was 0.65% at 80 mm rainfall and 0.79% at 100 mm. These figures indicate that areas with higher rainfall during the coldest quarter were favourable for wheat cultivation, with the probabilities approaching 0.80%, reflecting a strong potential for successful wheat farming (Fig 9e).

As precipitation exceeded 100 mm, the probability of wheat suitability gradually increased and peaked at 120 mm. For instance, the probability was 0.85% at 110 mm rainfall and 0.90% at 120 mm, indicating that regions with moderate to high rainfalls during the coldest quarter provided optimal conditions for wheat cultivation, with probabilities nearing maximum.

The probability of wheat suitability remained high (over 0.90%) after the precipitation levels exceeded 120 mm. At 130 mm rainfall, the probability increased to 0.94%, and at 132 mm, it was 0.95% (Fig 9e). The plateau indicates that areas receiving abundant rainfalls during the coldest quarter provided near-ideal conditions for wheat cultivation, with minimal variation in suitability after the rainfall surpasses 120 mm [70].

## 3.9 Future research

Future research could leverage advanced suitability, simulation and predictive models and consider, among others, cultural practices, socioeconomic influences, ecological elements and meteorological characteristics. These models could identify the optimal feature combination for determining optimal wheat cultivation sites and forecast wheat distribution under current conditions to guide management and resource allocation. Wheat production in arid and semi-arid environments should adopt Concepts of Agriculture 4.0 using the vast dataset generated by the Internet of Things (IoT), satellite imagery and agroclimatic models encompassing agronomic management and land use. Irrigation management, such as water deficit and water use efficiency, are critical water management factors in wheat production [71–74]. Furthermore, it is crucial for Oman to implement breeding programs that utilize genetic resources of climate-resilient wheat varieties, including local landraces adapted to high temperatures and drought conditions, as well as varieties selected by the Ministry of Agriculture, Fisheries Wealth, and Water Resources.

The analysis of five climatic variables and wheat distribution, encompassing both historical and projected data, reveals similar trends. The wheat distribution in both scenarios indicated a higher probability of wheat distribution when temperature seasonality was minimal. The probability of wheat distribution is elevated when the mean temperature ranges between 5 and 15 °C, which may physiologically enhance the flowering (anthesis) and grain-filling stages of wheat. Conversely, precipitation during the wettest months, typically occurring in the Al Hajar Mountains, where wheat has historically been distributed, involves occasional heavy rainfall from convective thunderstorms in spring and summer (March–May).

This precipitation may adversely affect the maturity and harvest stages of wheat production. Additionally, precipitation during the coldest quarter, encompassing December, January, and February, increases the probability of wheat production, coinciding with the physiological stages of seedling, tillering, stem elongation, and heading of wheat.

### 3.10. Implication and limitation of the modelling of technique

The outcomes of this study are vital for formulating agricultural policies and plans that are resilient to climate change. The research provides important insights for improving land-use strategies, fostering sustainable wheat farming, and supporting decision-making in the face of climate change. By identifying future agricultural suitability areas, this study helps in developing innovative strategies to enhance food security and ensure the sustainability of agriculture in semi-arid and arid regions.

The limitations are rooted in its reliance on presence-only data, which can compromise the representativeness of environmental conditions. It is also vulnerable to the selection and collinearity of predictor variables and presumes environmental stationarity, overlooking potential advancements in crop breeding and agricultural management. Moreover, uncertainties in climate projections, particularly those related to emission scenarios and downscaled climate models, may impact the reliability of long-term suitability assessments [75,76].

## 4. Conclusion

This study highlighted the intricate relationships between environmental factors and wheat cultivation and provided valuable insights into how climate variability influences crop suitability. Using MaxEnt modelling, the study determined that wheat thrives in regions with stable temperature seasonality, moderate cold season temperatures (below 12°C), and low to moderate rainfalls. However, wheat suitability declined when temperature seasonality exceeded 300, cold season temperatures surpassed 17°C–18°C, or precipitation during the coldest quarter exceeded 100 mm. Similarly, excessive rainfalls in the wettest month (above 75 mm) adversely affected wheat viability. Crucially, wheat demonstrated greater adaptability in areas with higher precipitation seasonality, likely due to better water availability during critical growth stages. These findings have significant implications for agricultural planning, climate adaptation strategies and the development of climate-resilient wheat varieties. Future research should investigate soil characteristics, land-use patterns and long-term climate projections to refine predictive models and establish a more robust framework for climate-resilient agriculture.

### Supporting information

**S1 File. S1.** Wheat distribution and inventory data from the most recent Agricultural Census published by the Ministry of Agriculture, Fisheries Wealth and Water Resources of Oman (https://www.mafwr.gov.om/). **S2.** Bioclimatic datasets in raster formats were utilised to model the spatial distribution of wheat in Oman for the period from 1970 to 2020. **S3.** Bioclimate datasets in raster formats were used to model and predict the future geographical distribution of wheat in Oman for the period 2021–2040. **S4.** Bioclimate datasets in raster formats were used to model and predict the future geographical distribution of wheat in Oman for the period 2040–2061.
(ZIP)

**S2 File. Bioclimate datasets in raster formats were used to model and predict the future geographical distribution of wheat in Oman for the period 2061–2080.**
(PDF)

### Acknowledgments

The authors gratefully thank The Ministry of Agriculture Fisheries Wealth & Water Resources for providing the facilities for supporting the related research.

## Author contributions

**Conceptualization:** Khalifa M. Al-Kindi, Ali Hussain Al-Lawati.

**Data curation:** Khalifa M. Al-Kindi, Ali Hussain Al-Lawati.

**Formal analysis:** Khalifa M. Al-Kindi.

**Funding acquisition:** Khalifa M. Al-Kindi, Ali Hussain Al-Lawati.

**Investigation:** Khalifa M. Al-Kindi, Ali Hussain Al-Lawati.

**Methodology:** Khalifa M. Al-Kindi, Ali Hussain Al-Lawati.

**Project administration:** Khalifa M. Al-Kindi, Ali Hussain Al-Lawati.

**Resources:** Khalifa M. Al-Kindi, Ali Hussain Al-Lawati.

**Software:** Khalifa M. Al-Kindi, Ali Hussain Al-Lawati.

**Supervision:** Khalifa M. Al-Kindi, Ali Hussain Al-Lawati.

**Validation:** Khalifa M. Al-Kindi, Ali Hussain Al-Lawati.

**Visualization:** Khalifa M. Al-Kindi, Ali Hussain Al-Lawati.

**Writing – original draft:** Khalifa M. Al-Kindi, Ali Hussain Al-Lawati.

**Writing – review & editing:** Khalifa M. Al-Kindi, Ali Hussain Al-Lawati.

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
