## [Decision Letter · Decision Letter 0]

PONE-D-24-57793Climate Change and its Impact on Wheat Distribution in Semi-arid EcosystemsPLOS ONE

Dear Dr. Alkindi,

Thank you for submitting your manuscript to PLOS ONE. After careful consideration, we feel that it has merit but does not fully meet PLOS ONE’s publication criteria as it currently stands. Therefore, we invite you to submit a revised version of the manuscript that addresses the points raised during the review process.

We look forward to receiving your revised manuscript.

Kind regards,

Randeep Singh

Academic Editor

PLOS ONE

3. We note that Figures 1 and 4 in your submission contain [map/satellite] images which may be copyrighted. All PLOS content is published under the Creative Commons Attribution License (CC BY 4.0), which means that the manuscript, images, and Supporting Information files will be freely available online, and any third party is permitted to access, download, copy, distribute, and use these materials in any way, even commercially, with proper attribution. For these reasons, we cannot publish previously copyrighted maps or satellite images created using proprietary data, such as Google software (Google Maps, Street View, and Earth). For more information, see our copyright guidelines: http://journals.plos.org/plosone/s/licenses-and-copyright.

1. You may seek permission from the original copyright holder of Figures 1 and 4 to publish the content specifically under the CC BY 4.0 license. 

Reviewers' comments:

Reviewer's Responses to Questions

**Comments to the Author**

1. Is the manuscript technically sound, and do the data support the conclusions?

Reviewer #1: Partly

Reviewer #2: Yes

2. Has the statistical analysis been performed appropriately and rigorously? 

Reviewer #1: Yes

Reviewer #2: Yes

3. Have the authors made all data underlying the findings in their manuscript fully available?

Reviewer #1: Yes

Reviewer #2: Yes

4. Is the manuscript presented in an intelligible fashion and written in standard English?

Reviewer #1: Yes

Reviewer #2: Yes

5. Review Comments to the Author

Reviewer #1: the manuscript topic is interesting and worth researching. the manuscript is well-organized and easy to follow. the methods are valid and reliable. however, the authors may consider improving the manuscript to enhance readership. i suggest a major revision with below specific comments:

- The research objectives would benefit from being more clearly defined to provide a stronger foundation for the reader.

- the significance of the study may be stated more clearly.

-Some of the figures lack clarity, and the resolution could be improved, particularly for those that contain detailed data or complex visualizations.

-besides, the caption of the figure could be more descriptive

- Please see the attachment

Reviewer #2: Overall, the authors succeeded in structuring and presenting their work in a scientifically rigorous manner. The proposed method combined well established research with elements of novelty.

The topic is interesting and important. However, several key areas need more work. I have summarized the required changes in the hope that the feedback will be useful to you as you update the paper.

1. (TITLE): The concise but keyword-rich title effectively anticipates the content and scope of the paper. Consider adding, tough, a reference to the Study area in the title itself, as it is the main novelty of the study results.

2. (ABST): The abstract provides all the information needed to understand the paper's main message and has a good logical flow. However, to better support the final claim, the authors could spend a few more words on the methodology and results instead of the concern with validating the MaxEnt model.

3. The authors highly recommended illustrating the methodology and the scenarios used in the study for the climate change models to project future climate data.

4. Proof reading by a native English speaker should be conducted to improve both language and organization quality.

5. The originality of the paper needs to be further clarified. Sufficient results are important to justify the novelty of a high-quality journal paper.

6. The authors do not explain the novelty of the present study compared to other studies. There is a need to identify the scientific gap that you come to cover and to state clearly how you cover it. Also, expect this fact, you have to highlight why your work is important to the field.

7. Please provide the paper with a flow chart showing the steps of conducting this study from beginning to end.

8. There is a strong need to explain the contribution of your work to the current research community.

9. (INTRODUCTION): In the Introduction, the authors succeed in narrowing down the context of their study, providing a valid ground for their objective.

• I suggest increasing the number of modern References studies which dealing with the core of the topic such as:

https://doi.org/10.1007/s40808-022-01358-x

https://doi.org/10.1504/IJGW.2021.115898

https://doi.org/10.1016/j.pce.2022.103224

10. I would ask you to discuss more about the present research's theoretical approach.

11. The quality of some figures is weak. The original (or editable) source of the figures should be used in the manuscript.

12. Much more explanations and interpretations should be added for the results, which are not enough.

13. It is suggested to compare the results of the present study with previous studies and analyze their results completely.

14. It is suggested to organize the conclusion section much better.

6. PLOS authors have the option to publish the peer review history of their article (what does this mean? ). If published, this will include your full peer review and any attached files.

**Do you want your identity to be public for this peer review?** For information about this choice, including consent withdrawal, please see our Privacy Policy .

Reviewer #1: No

Reviewer #2: No

---

## [Author Response · Author response to Decision Letter 1]

24 Feb 2025

Ref: PONE-D-24-57793

Title: Climate Change and its Impact on Wheat Distribution in Semi-arid Ecosystems

Journal: PLOS One

Dear, Editor

Thank you for consideration of our manuscript. Below we lay out rebuttal of comments made by reviewers in blue. We also have made changes to the manuscript as outlined as track changes in edited document. A ‘clean’ version of the updated manuscript is also uploaded to reflect these changes.

Reviewers' comments:

Reviewer #1:

- The manuscript topic is interesting and worth researching. The manuscript is well organized and easy to follow. The methods are valid and reliable. However, the authors may consider improving the manuscript to enhance readership. I suggest a major revision with the following specific comments:

Dear Reviewer,

We are deeply grateful for your favourable and productive comments regarding our manuscript. It is most encouraging to learn that you consider the subject matter engaging, find the document well-structured and easily digestible, and deem our methodological approach both sound and dependable.

- The research objectives would benefit from being more clearly defined to provide a stronger foundation for the reader.

Dear Reviewer,

We thank you for highlighting the need to clarify the research objectives. We agree that clearly defining objectives is essential to providing a strong foundation for the reader. In response, we have revised the introduction section to explicitly outline the research objectives. Please see lines 14-19 and 75-86 in the revised manuscript.

- The significance of the study may be stated more clearly.

Dear Reviewer,

Thank you for this feedback. Based on this comment, sentences have been added to the revised manuscript. In the introduction section, please see lines 55-81.

- Some of the figures lack clarity, and the resolution could be improved, particularly for those that contain detailed data or complex visualizations. Besides, the caption of the figure could be more descriptive. Please see the attachment.

We thank the reviewer for pointing out the need to improve the clarity, resolution, and descriptive quality of the figures and captions. We recognize the importance of presenting clear and high-quality visualizations to enhance the reader's understanding of the data. In response to your comment, we have made the following changes:

All figures, especially those containing detailed data or complex visualizations, have been updated to higher resolution formats to ensure clarity and readability, even when zoomed in or printed. Where necessary, we have adjusted the layout, font sizes, and color schemes of the figures to enhance their visual appeal and ensure the data is presented more effectively. Captions for all figures have been revised to provide more detailed descriptions, including explanations of the key elements and their relevance to the study. We carefully reviewed the attachment and addressed all specific points raised regarding individual figures. Any specific issues highlighted in the attachment have been rectified to align with your recommendations.

Reviewer #2:

- Overall, the authors succeeded in structuring and presenting their work in a scientifically rigorous manner. The proposed method combined well established research with elements of novelty. The topic is interesting and important. However, several key areas need more work. I have summarized the required changes in the hope that the feedback will be useful to you as you update the paper.

We sincerely thank the reviewer for their positive and encouraging feedback. We are delighted to hear that you found our manuscript to be scientifically rigorous, well-structured, and effectively presented. We appreciate your acknowledgment of the novelty in our proposed method and the importance of the topic. Your encouraging comments reinforce the relevance of our research and motivate us to continue refining and expanding this work. Thank you once again for recognizing the value of our study and for your thoughtful review.

- (TITLE): The concise but keyword-rich title effectively anticipates the content and scope of the paper. Consider adding, though, a reference to the Study area in the title itself, as it is the main novelty of the study results.

Dear Reviewer,

We appreciate your feedback. In response, we have modified the title to better reflect the research area and highlight the uniqueness of our findings. The revised title now reads: "Climate Change and its Impact on Wheat Distribution in Semi-Arid Ecosystems: A Case Study from Oman". This change can be found on lines 1-2 of the updated manuscript.

- (ABST): The abstract provides all the information needed to understand the paper's main message and has a good logical flow. However, to better support the final claim, the authors could spend a few more words on the methodology and results instead of the concern with validating the MaxEnt model.

Dear Reviewer,

We sincerely appreciate the reviewer’s feedback regarding the abstract and its logical flow. In response to the suggestion, we have revised the abstract to provide a more balanced presentation of the methodology and key results, while still addressing the importance of validating the MaxEnt model. Specifically, we have included a concise summary of the methodology, highlighting the use of bioclimatic variables and the MaxEnt model to predict the spatial distribution of wheat cultivation suitability. Additionally, we have briefly outlined the primary findings, such as the significant relationship between temperature seasonality and wheat probability, as well as the percentage of total suitable areas identified. These adjustments aim to ensure that the final claim is more strongly supported by a succinct overview of the methodology and results. Please see the abstract section in the revised manuscript.

- The authors highly recommended illustrating the methodology and the scenarios used in the study for the climate change models to project future climate data.

We appreciate the reviewer’s insightful comments and the suggestion to illustrate the methodology and scenarios used for projecting future climate data. In our study, we utilized the (CMIP6) framework for the climate changes models, which provides advanced simulations of climate conditions. A couple of words have been added to the revised manuscript. Please see lines 120-134, 138-144, 146-153, 157-161, and 171-175.

- Proof reading by a native English speaker should be conducted to improve both language and organization quality.

Dear Reviewer,

Thank you for your valuable feedback. We appreciate your suggestion regarding the language and organization of the manuscript. To address this, we have thoroughly proofread the manuscript to enhance its clarity, coherence, and overall language quality. Additionally, we have revised the structure where necessary to improve the flow and organization of the content. We believe these changes have significantly strengthened the manuscript, and we hope it now meets the expected standards.

- The originality of the paper needs to be further clarified. Sufficient results are important to justify the novelty of a high-quality journal paper.

We appreciate the reviewer’s valuable feedback and their suggestion to clarify the originality of our study and provide sufficient results to justify its novelty. In response, changes have been added to the revised manuscript. Please see lines 120-134, 138-144, 146-153, 157-161, and 171-175.

- The authors do not explain the novelty of the present study compared to other studies. There is a need to identify the scientific gap that you come to cover and to state clearly how you cover it. Also, except for this fact, you have to highlight why your work is important to the field.

Done. Please see lines 57-65, 66-72, 73-83, and 84-95.

- Please provide the paper with a flow chart showing the steps of conducting this study from beginning to end.

Done. Please see lines 172-176.

- There is a strong need to explain the contribution of your work to the current research community.

Thank you for your valuable feedback.

Firstly, our paper addresses a critical gap in the understanding of how climate change affects wheat distribution in semi-arid ecosystems, particularly in the Sultanate of Oman, where such localized studies are scarce. Please see lines 57-65 and lines 66-72.

Secondly, the study employs advanced modeling techniques, notably the Maximum Entropy (MaxEnt) model, to predict changes in suitable areas for wheat cultivation under different climate scenarios. This methodological approach enhances the precision of distribution predictions and offers a robust framework that can be adapted for similar studies in other semi-arid regions. Please see lines 73-83.

Moreover, the research contributes to the broader discourse on food security in the context of climate change. By identifying potential shifts in wheat cultivation zones, it provides valuable information for policymakers, agricultural planners, and farmers to develop adaptive strategies that ensure sustainable food production despite changing climatic conditions. Please see the results and discussion sections.

Lastly, this study integrates bioclimatic data from 1970 to 2020, offering a comprehensive temporal analysis that enriches the understanding of long-term climate trends and their agricultural impacts. This historical perspective, combined with future projections, makes the research highly relevant to ongoing discussions about climate resilience in agriculture.

In conclusion, this work not only fills a regional research gap but also contributes methodologically and practically to the global effort of understanding and mitigating the impacts of climate change on crop distribution.

- (INTRODUCTION): In the Introduction, the authors succeed in narrowing down the context of their study, providing valid ground for their objective. I suggest increasing the number of modern References studies which dealing with the core of the topic such as:

https://doi.org/10.1007/s40808-022-01358-x

https://doi.org/10.1504/IJGW.2021.115898

https://doi.org/10.1016/j.pce.2022.103224

I would ask you to discuss more about the present research's theoretical approach.

We appreciate the reviewer’s valuable suggestion to enhance the Introduction by incorporating more modern references relevant to the core of our study including the suggested sources above. In response, we have included the following recent studies to strengthen the theoretical foundation and contextual background of our research. Furthermore, we have expanded the discussion on the theoretical framework underpinning our study. We elaborated on the principles of spatial machine learning models, particularly Maximum Entropy (MaxEnt). This theoretical elaboration enhances the clarity of how our study fits within the broader context of groundwater potential assessment and predictive modeling in semi-arid regions.

- The quality of some figures is weak. The original (or editable) source of the figures should be used in the manuscript.

Dear Reviewer,

Thank you for your feedback,

All figures, especially those containing detailed data or complex visualizations, have been updated to higher resolution formats to ensure clarity and readability, even when zoomed in or printed. Where necessary, we have adjusted the layout, font sizes, and color schemes of the figures to enhance their visual appeal and ensure the data is presented more effectively. Captions for all figures have been revised to provide more detailed descriptions, including explanations of the key elements and their relevance to the study. We carefully reviewed the attachment and addressed all specific points raised regarding individual figures. Any specific issues highlighted in the attachment have been rectified to align with your recommendations.

- Much more explanations and interpretations should be added for the results, which are not enough. It is suggested to compare the results of the present study with previous studies and analyze their results completely.

We sincerely appreciate the reviewer’s insightful feedback highlighting the need for more detailed explanations and interpretations of the results. In response, we have substantially revised the Results and Discussion section to enhance its clarity, depth, and interpretative value. Please see lines 122, 269-270, 275-278, 308-215, 562-580, and 617-624.

- It is suggested to organize the conclusion section much better.

We thank the reviewer for the constructive suggestion to improve the organization of the conclusion section. In response, we have thoroughly revised this section to enhance its clarity, coherence, and overall structure. Please see lines 626-638 in the revised manuscript.

---

## [Decision Letter · Decision Letter 1]

PONE-D-24-57793R1Climate Change and its Impact on Wheat Distribution in Semi-arid Ecosystems: A case study from the Sultanate of OmanPLOS ONE

Dear Dr. Khalifa,

Thank you for submitting your manuscript to PLOS ONE. After careful consideration, we feel that it has merit but does not fully meet PLOS ONE’s publication criteria as it currently stands. Therefore, we invite you to submit a revised version of the manuscript that addresses the points raised during the review process.

Both the reviewers suggested some changes, kindly revise the manuscript and submit again.

We look forward to receiving your revised manuscript.

Kind regards,

Randeep Singh

Academic Editor

PLOS ONE

Reviewers' comments:

Reviewer's Responses to Questions

**Comments to the Author**

1. If the authors have adequately addressed your comments raised in a previous round of review and you feel that this manuscript is now acceptable for publication, you may indicate that here to bypass the “Comments to the Author” section, enter your conflict of interest statement in the “Confidential to Editor” section, and submit your "Accept" recommendation.

Reviewer #2: All comments have been addressed

Reviewer #3: (No Response)

Reviewer #4: (No Response)

2. Is the manuscript technically sound, and do the data support the conclusions?

Reviewer #2: Yes

Reviewer #3: Yes

Reviewer #4: Partly

3. Has the statistical analysis been performed appropriately and rigorously? 

Reviewer #2: Yes

Reviewer #3: Yes

Reviewer #4: I Don't Know

4. Have the authors made all data underlying the findings in their manuscript fully available?

Reviewer #2: Yes

Reviewer #3: Yes

Reviewer #4: No

5. Is the manuscript presented in an intelligible fashion and written in standard English?

Reviewer #2: Yes

Reviewer #3: Yes

Reviewer #4: Yes

6. Review Comments to the Author

Reviewer #2: After a re-review, I'm satisfied with the authors' responses. The revised manuscript and the authors' letter address my primary concerns. I have no other comments and I have enjoyed reading the revised manuscript. I found the manuscript worthy of publication. Therefore, I recommend publishing this paper in the current form. I thank the authors and congratulate them.

Reviewer #3: "Climate Change and its Impact on Wheat Distribution in Semi-arid Ecosystems: A case study from the Sultanate of Oman" This study investigates the relationships between wheat distribution and a set of predetermined climatic factors (for both current and future scenarios). After reviewing the revised manuscript and the authors' responses to the reviewers' comments, the work appears well-designed in its current form, with results presented clearly and logically. The study is now recommended for acceptance pending the incorporation of the following changes.

Discussion:

If the journal’s guidelines permit, I suggest tidying up the Discussion section by separating it into an independent section. Within this section, you could describe the relative contributions of individual climatic variables to the target species’ distribution. Additionally, I recommend including two subsections, addressing the limitations and implications of the modeling technique. For this, the following references conducted in the Middle East may be relevant. In any case, adding few sentences on the limitations and implications of the study is required.

https://doi.org/10.1016/j.ecoinf.2022.101930

https://doi.org/10.1007/s10661-024-12438-z

https://www.mdpi.com/2071-1050/14/21/14621

https://doi.org/10.1016/j.jnc.2023.126505

Reviewer #4: This study investigates the influence of temperature and precipitation on wheat distribution in the Sultanate of Oman, classifying regions into varying suitability levels. The findings offer significant insights and practical guidance for wheat cultivation in the area. However, several issues need to be addressed, as outlined below:

1. The abstract requires substantial refinement and should be rephrased to align more closely with the central theme and key findings of the study. For instance, the first sentence of the abstract is identical to the opening sentence of the introduction, which is not permissible in academic writing. The abstract should succinctly encapsulate the primary objectives, methodologies, results, and conclusions of the research.

2. The WorldClim database comprises 19 bioclimatic variables. A clear rationale and justification for the selection of the 9 variables used in this study, as well as the exclusion of the remaining 10, should be provided. This will enhance transparency regarding the criteria employed in the variable selection process.

3. Topographic characteristics, hydrological distribution, and soil properties are critical determinants of plant growth and distribution. It is essential to explain why these factors were not incorporated into the model. Their inclusion could offer a more holistic assessment of wheat suitability and improve the robustness of the findings.

4. The classification of suitability into five categories—extremely high, high, medium, low, and extremely low—requires clarification. Please provide the specific criteria or methodological framework used to define these categories. This will facilitate a clearer understanding of how the suitability thresholds were established and interpreted within the context of the study.

7. PLOS authors have the option to publish the peer review history of their article (what does this mean? ). If published, this will include your full peer review and any attached files.

**Do you want your identity to be public for this peer review?** For information about this choice, including consent withdrawal, please see our Privacy Policy .

Reviewer #2: No

Reviewer #3: No

Reviewer #4: No

---

## [Author Response · Author response to Decision Letter 2]

20 Mar 2025

Dear Editor and Reviewer’s,

We wish to convey our heartfelt gratitude for the detailed and insightful feedback on our manuscript. We are genuinely thankful for the time and effort you dedicated to offering such comprehensive insights. Below, you will find our responses to your comments, along with the revisions we have made to improve the clarity, precision, and originality of our research.

Regards,

Al-Kindi

Review Comments to the Author

Reviewer #2: After a re-review, I'm satisfied with the authors' responses. The revised manuscript and the authors' letter address my primary concerns. I have no other comments and I have enjoyed reading the revised manuscript. I found the manuscript worthy of publication. Therefore, I recommend publishing this paper in the current form. I thank the authors and congratulate them.

We sincerely appreciate the time and effort you invested in reviewing our manuscript. Your constructive feedback has been crucial in improving the quality of our work. We are delighted to learn that you found our revisions satisfactory and that our responses adequately addressed your primary concerns. Your positive comments and recommendation for publication are incredibly motivating. Thank you once again for your thoughtful review and kind words. We greatly value your support and significant contribution to this research.

Reviewer #3: "Climate Change and its Impact on Wheat Distribution in Semi-arid Ecosystems: A case study from the Sultanate of Oman" This study investigates the relationships between wheat distribution and a set of predetermined climatic factors (for both current and future scenarios). After reviewing the revised manuscript and the authors' responses to the reviewers' comments, the work appears well-designed in its current form, with results presented clearly and logically. The study is now recommended for acceptance pending the incorporation of the following changes.

Dear Reviewer,

We are truly grateful for your insightful review and the positive remarks regarding our manuscript. It is gratifying to know that you consider the study to be well-structured and that the revisions have enhanced the clarity and logical flow of the results. We recognize your suggestions for additional modifications and will thoughtfully integrate them to further improve the manuscript's quality. Your valuable insights have significantly contributed to the refinement of our work, and we appreciate the time and effort you have dedicated to reviewing our study.

Discussion:

If the journal’s guidelines permit, I suggest tidying up the Discussion section by separating it into an independent section. Within this section, you could describe the relative contributions of individual climatic variables to the target species’ distribution. Additionally, I recommend including two subsections, addressing the limitations and implications of the modeling technique. For this, the following references conducted in the Middle East may be relevant. In any case, adding few sentences on the limitations and implications of the study is required.

https://doi.org/10.1016/j.ecoinf.2022.101930

https://doi.org/10.1007/s10661-024-12438-z

https://www.mdpi.com/2071-1050/14/21/14621

https://doi.org/10.1016/j.jnc.2023.126505

Dear Reviewer,

We value your feedback. Based on your comments and suggestions, we have incorporated sentences into the revised manuscript that clarify the relative contributions of individual climate variables to the distribution of the target species. Please refer to lines 633-644. Furthermore, we have added a new section to the discussion, which explores the implications and limitations of the modeling technique. See lines 645-657 for this addition. Consequently, we have included more pertinent citations in the revised manuscript.

Reviewer #4: This study investigates the influence of temperature and precipitation on wheat distribution in the Sultanate of Oman, classifying regions into varying suitability levels. The findings offer significant insights and practical guidance for wheat cultivation in the area. However, several issues need to be addressed, as outlined below:

1. The abstract requires substantial refinement and should be rephrased to align more closely with the central theme and key findings of the study. For instance, the first sentence of the abstract is identical to the opening sentence of the introduction, which is not permissible in academic writing. The abstract should succinctly encapsulate the primary objectives, methodologies, results, and conclusions of the research.

Thank you very much for your feedback. We have updated the abstract in the revised version of the manuscript. Please see the abstract section in the revised version.

2. The WorldClim database comprises 19 bioclimatic variables. A clear rationale and justification for the selection of the 9 variables used in this study, as well as the exclusion of the remaining 10, should be provided. This will enhance transparency regarding the criteria employed in the variable selection process.

Dear Reviewer,

We appreciate your feedback. We have deliberated on the choice of the WorldClim database. Please refer to lines 143-149. Additionally, new sentences have been incorporated into the updated manuscript. Please check lines 137-139.

3. Topographic characteristics, hydrological distribution, and soil properties are critical determinants of plant growth and distribution. It is essential to explain why these factors were not incorporated into the model. Their inclusion could offer a more holistic assessment of wheat suitability and improve the robustness of the findings.

Dear Reviewer,

We appreciate your comment. We acknowledge that terrain features, water resource distribution, and soil properties are key factors affecting plant growth and distribution. These aspects influence water availability, nutrient dynamics, and microclimatic conditions, all of which are essential for wheat cultivation. Elements such as elevation, slope, and aspect impact temperature variations and soil moisture retention, while hydrological factors like groundwater access and drainage patterns are crucial for sustaining crop productivity. Furthermore, soil characteristics, including texture, organic matter content, and pH, directly affect root development and nutrient uptake. Although these factors are highly relevant for evaluating wheat suitability, they were not directly included in our modeling approach, which focused primarily on climatic variables due to their dominant influence on crop suitability in future climate scenarios. Given the regional scope of our analysis and the limitations in the spatial resolution of certain soil and hydrological datasets, we prioritized climate-based predictors to ensure consistency in the model's application. Nonetheless, recognizing their significance, we recommend that future research incorporate these additional parameters to provide a more comprehensive assessment of wheat cultivation potential. This information has already been mentioned in our manuscript. Please refer to the Future Research Section (lines 620-631).

4. The classification of suitability into five categories—extremely high, high, medium, low, and extremely low—requires clarification. Please provide the specific criteria or methodological framework used to define these categories. This will facilitate a clearer understanding of how the suitability thresholds were established and interpreted within the context of the study.

Dear Reviewer,

We sincerely value your input. We have added several sentences to the revised manuscript, which can be found on lines 176-184 and lines 236-238.

---

## [Editor Report · Decision Letter 2]

Climate Change and its Impact on Wheat Distribution in Semi-arid Ecosystems: A case study from the Sultanate of Oman

PONE-D-24-57793R2

Dear Dr. Khalifa,

We’re pleased to inform you that your manuscript has been judged scientifically suitable for publication and will be formally accepted for publication once it meets all outstanding technical requirements.

Kind regards,

Randeep Singh

Academic Editor

PLOS ONE
---

## [Editor Report · Acceptance letter]

PONE-D-24-57793R2

PLOS ONE

Dear Dr. Alkindi,

I'm pleased to inform you that your manuscript has been deemed suitable for publication in PLOS ONE. Congratulations! Your manuscript is now being handed over to our production team.

Kind regards,

on behalf of

Dr. Randeep Singh

Academic Editor

PLOS ONE